# The Importance of Physical Activity to Augment Mood during COVID-19 Lockdown

**DOI:** 10.3390/ijerph19031270

**Published:** 2022-01-24

**Authors:** Curtis Fennell, Thomas Eremus, Moisés Grimaldi Puyana, Borja Sañudo

**Affiliations:** 1Exercise and Nutrition Science Program, University of Montevallo, Montevallo, AL 35115, USA; teremus@forum.montevallo.edu; 2Department of Physical Education and Sport, Universidad de Sevilla, 41013 Seville, Spain; mgrimaldi@us.es (M.G.P.); bsancor@us.es (B.S.)

**Keywords:** COVID-19, mental health, physical health, POMS

## Abstract

The purpose of this study was to assess the effect of COVID-19 lockdown on mood and objective physical activity. A sample of 78 college students in Spain completed an assessment of mood using the valid Profile of Mood State (POMS) questionnaire and had their physical activity tracked objectively using a validated wrist-worn accelerometer (Xiaomi Mi Band 2) for one week before being under COVID-19 lockdown (T1) and for one week during COVID-19 lockdown (T2). Paired samples *t*-tests revealed significantly greater (*p* = 0.027) POMS Total Mood Disturbance (TMD) Score T2 (mean ± SD) (22.6 ± 28.0) compared to T1 (17.7 ± 22.6) (lower score represents better mood) and significantly lower (*p* ≤ 0.001) POMS Vigor Score T2 (14.1 ± 5.0) compared to T1 (18.2 ± 4.5) (lower vigor score represents lower mental and physical energy levels). Additionally, Total Objective Steps was significantly less (*p* ≤ 0.001) lT2 (15,841.9 ± 17,253.2 steps) compared to T1 (64,607.0 ± 50,525.2 steps). Regression analyses demonstrated significant negative relationships of Total Objective Steps and POMS Depression (*p* = 0.014, Beta = −0.277, *t* = −2.511), POMS Anger (*p* = 0.040, Beta = −0.233, *t* = −2.091), and POMS TMD (*p* = 0.007, Beta = −0.302, *t* = −2.754) T2. The regressions also revealed a significant positive relationship (*p* = 0.012, Beta = 0.283, *t* = 2.57) of Total Objective Steps and POMS Vigor T2. These data suggest that being in a lockdown due to a pandemic may have negative physical and mental health-related consequences and that engaging in physical activity may reduce these deleterious mental health-related consequences during lockdowns and quarantines.

## 1. Introduction

During the recent global pandemic, the World Health Organization (WHO) advised a worldwide lockdown to minimize the spread of the coronavirus (COVID-19). Spain imposed a lockdown on 14 March 2020; thus, its citizens were forced to stay lockdown at home except for “essential tasks” (e.g., grocery shopping, buying medication, assisting the sick or disabled, etc.). Consequently, indoor and outdoor commercial and municipal activities were suspended, including access to fitness facilities, parks, sidewalks, etc. [1,2,3,4,5,6,7].

Being in a lockdown environment (e.g., mandate to reduce time outside of home to slow down the spread of the infection) due to a lockdown may alter the activities in which individuals participate each day due to rapid changes in work and social environments, which may have a large impact on behavior and mood [8,9]. Research on the impact of COVID-19 lockdown and quarantine on mood found this resulted in negative psychological effects [10,11,12,13]. Activities that may have changed due to decreased time away from home include decreased time engaging in physical activity and increased sedentary activity time [1]. These activities have been independently shown to affect mood state, as sedentary behavior is negatively associated with mood and physical activity is positively associated with mood [14,15,16,17,18,19,20,21,22]. During the COVID-19 lockdown, previous studies have shown that sedentary activity has increased [1,2] whilst physical activity decreased [1,2,4,5,6,7,23]. Therefore, mood could also be decreased due to the lockdown because of decreased physical activity; but this has yet to be fully explored.

Prior to the COVID-19 pandemic, there was already a global concern regarding the high rates of anger, depression, suicide, etc. [24]. During the pandemic, studies have confirmed significant mood disturbances such as elevated tension, depression, anger, confusion, fatigue, and reduced vigor [25,26,27,28], especially those aged ≤ 25 years [26]. A large percentage of the population have experienced difficult emotional adjustments due to extended periods of confinement, deaths, and increased social adversity that may lead to depression, low mood, irritability, stress, etc., during lockdowns [28]. Since physical activity enhances mood and sedentary behavior decreases mood, having less access to physical activity options (e.g., parks, fitness facilities, exercise equipment, etc.) during lockdown, could lead to decreased mood [6]. The lockdown is put in place to keep individuals healthy. However, if the lockdown is contributing to individuals engaging in less mood- and health-enhancing activities, including physical activity, then the detriments to health elicited by the lockdown may be more deleterious to health than the benefits of the lockdown. Additionally, cessation of physical activity may incur metabolic/cardiovascular disturbances that promote obesity, impaired insulin sensitivity, and lipid metabolism [29]. Sedentary behavior and lack of physical activity associated deaths are increasing in adults [30,31]. Because there was a change in lifestyle due to the COVID-19 pandemic, and healthy activities are crucial to preventing disease, it is useful to study how the COVID-19 lockdown affected individual’s health-related lifestyle choices (e.g., physical activity) and mental health parameters (e.g., mood) [32].

Therefore, the purpose of this research was to assess the effect of the COVID-19 pandemic lockdown on objectively measured physical activity and subjectively measured mood and to assess the relationship of mood and physical activity during COVID-19 lockdown in college-aged students in Spain. Due to the population having to spend greater time at home and having less access to fitness facilities, sidewalks, parks, etc. during the lockdown, our hypotheses were as follows: Hypothesis 1 (H1): the COVID-19 pandemic lockdown would result in decreased objective physical activity; Hypothesis 2 (H2): the lockdown would result in decreased mood; Hypothesis 3 (H3): objective physical activity would predict increased mood (i.e., positive association with mood); Hypothesis 4 (H4): objective physical activity would negatively predict depression (i.e., inversely associated with depression).

## 2. Materials and Methods

### 2.1. Participants

This study was conducted at the University of Sevilla in Spain. For data collection, participants were recruited prior to the COVID-19 lockdown. Recruitment was done by distributing an invitation through administrative channels and snowball sampling through social media to about 5000 students from five universities in Spain. Participants were college students in Seville, Spain who were 18–36 years old. The study was conducted according to the Declaration of Helsinki, was approved by the University of Sevilla ethical committee, and all participants signed the informed consent. The protocol code for the study was 319-N-19 and the date of approval was 7 February 2019.

### 2.2. Protocol

There were two phases of this study. Phase 1 was conducted throughout a single week prior to the COVID-19 lockdown (T1) in February 2020. On day 1 of phase 1, participants were given a validated physical activity monitor to wear on their wrist for one week. The participants were instructed to wear the wristband as much as possible throughout the day except when he or she was submerged in water or taking a shower. Participants who did not wear the accelerometer for at least 10 h during waking time each day were removed from the study. Data from the wristband accelerometer were automatically uploaded to a computer database for collection that only the researchers could view. At the end of the week, participants completed a virtual survey of their mood from the previous week.

Phase 2 was conducted throughout a single week during the COVID-19 lockdown (T2), in March and April 2020. Here, participants repeated the protocol in phase 1. Researchers contacted participants to ensure that their wristband accelerometers were working correctly and assisted the participants in completing the questionnaire remotely after the last day of the week. Data were collected once again after completion.

### 2.3. Measurements

The data collected included, sex, age, education, occupation, steps (physical activity), and mood. To assess objective physical activity, a wristband accelerometer that has been reported as valid and reliable for the purposes of measuring heart rate, number of steps, and distance [33,34] was utilized. Participants wore the Xiaomi Mi Band 2 wrist-worn accelerometer (Beijing, China) which objectively measures the number of steps moved each day. Steps recorded from the Xiaomi Mi Band 2 was used to assess objective physical activity. To assess mood, the Profile of Mood States (POMS) [35] questionnaire was used. The POMS is a self-report questionnaire that assesses how the individual has been feeling the past week, including the current day. It consists of 58 items on a 5-point Likert scale from 0 (not at all) to 4 (extremely). The items are grouped into six subscales: Depression (15 items, range 0–60), Fatigue (7 items, range 0–28), Tension (9 items, range 0–36), Confusion (7 items, range 0–28), Anger (12 items, range 0–48), and Vigor (8 items, range 0–32). The test also provides a Total Mood Disturbance (TMD) score, which is a summary of positive mood from adding subscales, and subtracting vigor (range 0–200). Higher scores for the TMD indicate a greater mood disturbance. The Cronbach-alpha reliability for the POMS was reported at 0.63 to 0.96. The total score for each scale was used as an outcome measure.

### 2.4. Statistical Analysis

Data were reported as means (±standard deviations) for continuous variables or percentages for categorical variables. To examine physical activity and mood differences induced by the lockdown, comparisons among T1 and T2 time-points were assessed using paired samples *t*-tests. To investigate the associations between physical activity and mood states, multiple linear regressions were conducted accounting for other covariates (e.g., sex, weight, and age). A *p*-value of ≤0.05 was deemed statistically significant. Statistical analyses were performed using IBM SPSS Statistics for Windows (Version 24.0, IBM Corp., Armonk, NY, USA).

## 3. Results

Of the 5000 students contacted for participation in the study, a total of 754 were willing to participate. Participants were excluded if they did not complete the baseline questionnaire, refused to participate in the study, refused to wear the accelerometer for at least 10 h per day, had a health problem that negatively impacted participation in physical activity (e.g., orthopedic injury), or used any medication that could affect mood (e.g., antidepressants, opioids, etc.) Due to this criteria, 119 students were assessed at baseline (65% males) and 78 valid surveys and accelerometer data were collected due to participant drop-out and incomplete data (e.g., did not wear the accelerometer for at least 10 h during waking time each day). Therefore, the sample size amounted to 78 participants, ages 18–36 years old. Participant characteristics are presented in Table 1. Results are reported as (mean ± SD).

### 3.1. Pre-Lockdown to During-Lockdown Differences

Table 2 includes the results of the paired samples *t*-tests. Participants exhibited significantly greater (*p* = 0.027) POMS TMD lockdownT2 (22.6 ± 28.0) compared to T1 (17.7 ± 22.6) (Figure 1). Additionally, participants demonstrated significantly less (*p* ≤ 0.001) POMS Vigor T2 (14.1 ± 5.0) compared to T1 (18.2 ± 4.5) (Figure 2). The total objectively measured steps per week was significantly less (*p* ≤ 0.001) T2 (15,841.9 ± 17,253.2 steps) compared to T1 (64,607.0 ± 50,525.2 steps) (Figure 3). There were no significant differences (p > 0.05) for any other of the POMS variables from T1 to T2.

### 3.2. Associations of Variables

Table 3 includes the results of the regression analyses. There was a significant negative relationship (*p* = 0.014, Beta = −0.277, *t* = −2.511) of Total Steps and POMS Depression T2. In other words, as physical activity increased, depression decreased during lockdown. There was a significant negative relationship (*p* = 0.040, Beta = −0.233, *t* = −2.091) of Total Steps and POMS Anger T2. In other words, as physical activity increased, anger decreased during lockdown. Furthermore, there was a significant positive relationship (*p* = 0.012, Beta = 0.283, *t* = 2.57) of Total Objective Steps and POMS Vigor T2. In other words, as physical activity increased, energy levels also increased during lockdown. Last, the data revealed a significant negative relationship (*p* = 0.007, Beta = −0.302, *t* = −2.754) of Total Steps and POMS TMD T2. A low score on the TMD measure for the POMS displays a good overall mood and a high score represents a poor overall mood. Therefore, these results reveal that, as physical activity increased, mood increased during lockdown. There were no relationships (*p* > 0.05) of Total Steps with any other of the POMS variables at both T1 and T2.

## 4. Discussion

The results indicate that, during lockdown due to the COVID-19 pandemic, participants’ mood and physical and mental energy decreased compared with pre-lockdown. Additionally, participants decreased their objectively assessed physical activity (i.e., number of steps) by an average of 75% during lockdown compared with pre-lockdown. Moreover, this investigation demonstrated negative associations of physical activity with depression and anger during lockdown and positive associations of physical activity with energy and mood during lockdown. This study emphasizes the importance of engaging in physical activity during pandemic lockdowns.

The decrease in physical activity during the COVID-19 pandemic demonstrated in the current study was in line with our first hypothesis (H1) and agreed with previous investigations [1,4,9,32,36,37,38,39,40,41,42]. The systematic review by Stockwell, et al., (2021), utilized 45 articles that assessed physical activity during COVID-19 in healthy adults, to conclude that all the studies reported total physical activity to decrease during COVID-19 [37]. Upon further research, two additional studies were identified that resulted in some subjects who increased their physical activity during COVID-19 [41,42]. However, Anyan, et al., (2020) separated the sample into reduced, unchanged, and increased physical activity groups; therefore, due to individual variation, there were a few subjects who increased their physical activity [41]. Furthermore, Peterson (2021) found no significant changes in physical activity over 8 weeks of COVID-19; however, of the four time points measured, time point three had the lowest proportion of the population classified as participating in sufficient physical activity [42]. Two studies found a decrease in participants physical activity only in the most active groups, whereas the moderate- and low-active groups increased their overall physical activity [2,6]. Overall, the current investigation agrees with most previous studies that, in healthy adults, during the COVID-19 lockdown, there was a reduction in overall physical activity [1,4,9,32,36,37,38,39,40,41,42]. The present investigation is important to add to the body of literature on this topic because, of the 66 total studies cited in the systematic review by Stockwell, et al., (2021), only four utilized objective measures of physical activity [1,38,39,40]. Therefore, objective measures of physical activity are warranted to reduce errors of subjective assessments such as user bias and difficulty with recall [43,44,45].

The present and previous investigations demonstrating that physical activity is reduced during COVID-19 lockdown does not come without potential detriments to health. There is extensive evidence that regularly engaging in physical activity elicits many benefits to physical health [17,18,19,20,29,30,46,47], mood, mental and physical energy and reduces negative mental health [48,49,50]. Additionally, physical inactivity is associated with an increase of developing cardiometabolic diseases, cancer, and premature mortality [51,52,53], while there is also a 32% increased risk of hospitalization from COVID-19 [54]. Moderate-intensity physical activity has acute benefits on immune function and inflammation that can help reduce the severity of COVID-19 outcomes and be protective to those with comorbidities [55,56]. Recently, Sallis, et al., 2021 assessed 48,440 COVID-19 patients’ subjective physical activity prior to being diagnosed with the virus to find those who met the physical activity guidelines were 70% less likely to be hospitalized and about 80% less likely to die [53]. Because the sample in the current investigation displayed a large decrease in objectively measured steps during lockdown, this could lead to a reduction in individual’s physical and mental health, and place someone at higher risk to suffer consequences from the COVID-19 virus.

In agreement with our second hypothesis (H2), this study revealed that the sample decreased mood during the lockdown in place due to the COVID-19 pandemic. In addition, the results demonstrated that participants had decreased energy during lockdown. We also hypothesized physical activity would predict increased mood (H3) and be inversely associated with depression (H4), which were both supported by the data. The regression analyses results indicated that during lockdown, physical activity predicted increased energy, decreased depression, decreased anger, and increased overall mood. Again, this highlights the importance of engaging in physical activity to prevent a decrease and mood and enhance individual’s mood when it is decreased during lockdown. Previous studies that assessed the effect of COVID-19 lockdown on mood parameters showed the lockdown led to negative psychological effects, such as decreased energy [42], increased depression [6,32,37,38,52] and decreased overall mood [6,10,11,12,36,41,42], which are in agreement with the present investigation. Furthermore, research has shown that the COVID-19 pandemic has elicited a cascade of negative psychological and behavioral effects that have been further exacerbated by personality traits, alexithymia, and resilience [25,57]. For example, Clemente-Suárez, (2020) demonstrated that, during the pandemic, a wide array of negative thoughts and emotions may be experienced, such as depression, anxiety, delirium, psychosis, sadness, fear, uncertainty, and even suicidality due the psychological and psychosocial stressors/triggers of the pandemic and being in isolation, which may be pronounced in individuals who present mental pathologies prior to the pandemic [25]. In addition to these negative wellbeing symptoms, research has also shown that decreased mood during the lockdown was associated with a poorer diet, suggesting emotional eating to occur [4,58,59].

An interesting point of the current research is we demonstrated physical activity predicted improving mental health. This was shown by the regressions, which showed a positive relationship between physical activity and energy and physical activity and mood during lockdown, while a negative relationship was found with physical activity and depression and physical activity and anger during lockdown. These findings are not surprising as they are consistent with recent research that shows that, during the COVID-19 lockdown, physical activity was associated with lower mental health problems [5,6,41,42,59,60] and reductions in levels of anxiety and depression [42]. Likewise, Anyan, et al., (2020) demonstrated that subjects who reduced physical activity during the COVID-19 lockdown had the highest levels of depression and those who increased physical activity reported the lowest levels of depression [41]. Additionally, Meyer, et al., (2020) showed associations between reduction in physical activity with higher negative mental health, lower positive mental health, and more severe anxiety and depressive symptoms during COVID-19 lockdown [6]. Moreover, López-Bueno, et al., (2020), demonstrated that those who adhered to physical activity guidelines during COVID-19 were associated with lower perceived anxiety and lower perceived worse mood [5]. Furthermore, Brand, et al., (2020), measured subjective well-being and exercise frequency to find those who exercised during the pandemic experienced better mood and those who reduced their exercise frequency during the pandemic reported the worst mood [60]. Overall, there is overwhelming evidence suggesting that engaging in physical activity during times of lockdown is drastically important to prevent decreases in mood and to enhance mood.

## 5. Limitations and Future Research

Although this is a novel study that assessed objective physical activity and subjective mood before and during the COVID-19 lockdown in college students in Spain, it is not without limitations. This study is limited to a small sample size of 78 college students at the University of Sevilla in Spain. Additionally, this was a non-experimental study, and therefore causal inference cannot be made. In other words, we cannot assume that the COVID-19 lockdown caused the reduced physical activity and mood during the lockdown. Therefore, future studies should incorporate large sample sizes that include all ages of individuals in differing communities and cultures to encompass the effect of lockdown and quarantine on objective physical activity and mood. Additionally, experimental studies should be conducted to assess the effect of lockdown on individuals’ physical activity and mood.

## 6. Conclusions

In summary, this investigation compared objective physical activity (i.e., steps) and subjective mood parameters for one week pre-lockdown and one week during lockdown from the COVID-19 pandemic. The lockdown resulted in lower mood and energy, while objectively measured steps were less. Regression analyses revealed physical activity predicted lower depression, lower anger, increased energy, and increased mood during lockdown. This study suggests the importance of participating in physical activity during lockdown because of the physical and mental benefits associated with engaging in regular physical activity. In conclusion, this study can be used to advise the development of policies and guidelines that aim at promoting physical activity through public health awareness messages about the importance to engage in regular physical activity to improve health-related variables and prevent decreases in mood and energy while individuals are under a lockdown or quarantine mandate.

## Figures and Tables

**Figure 1 ijerph-19-01270-f001:**
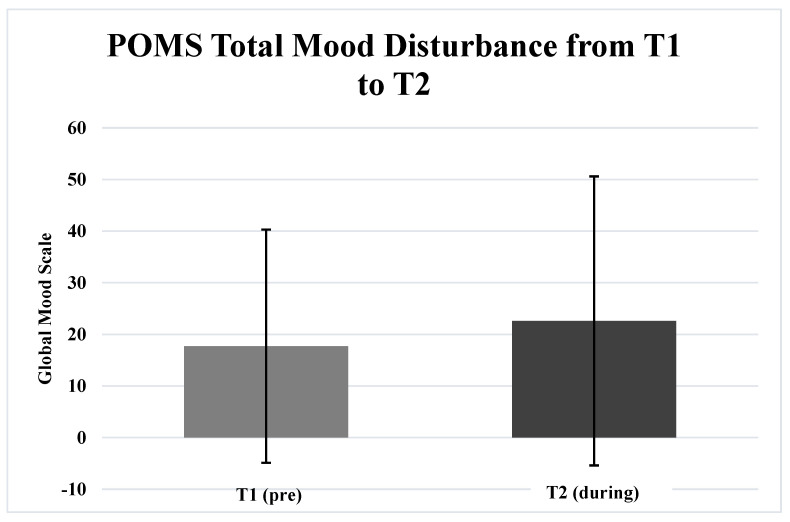
This figure illustrates the complete measure of mood measured by the POMS of participants one week before (T1) and one week during (T2).

**Figure 2 ijerph-19-01270-f002:**
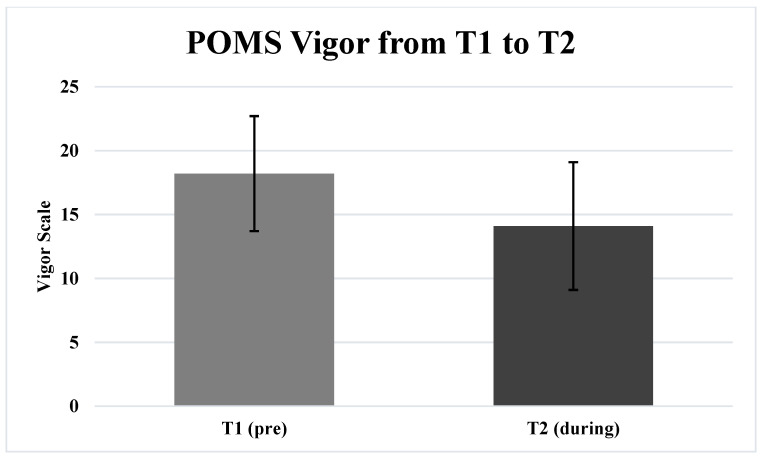
This figure depicts the energy levels (vigor) measured by the POMS in participants for one week before (T1) and one week during (T2).

**Figure 3 ijerph-19-01270-f003:**
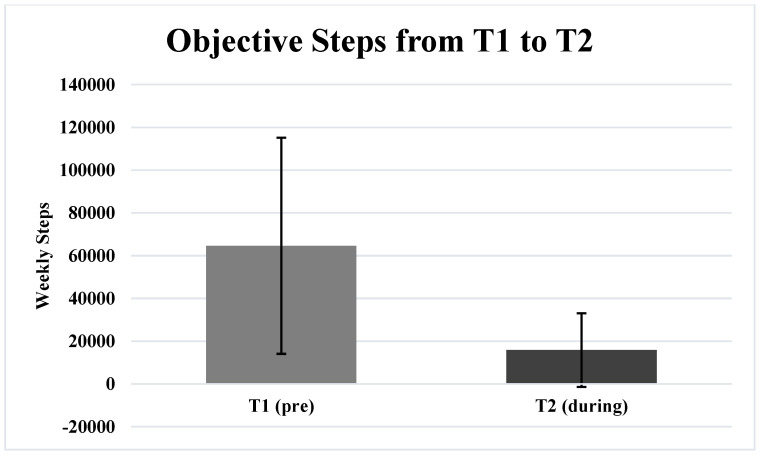
This illustration depicts the total objectively-measured steps for one week before (T1) and one week during (T2).

**Table 1 ijerph-19-01270-t001:** Participant Characteristics (*n* = 78).

Characteristic	N	%
Male	52 (22.2 ± 3.4 years old)	66.7
Female	26 (22.9 ± 2.5 years old)	33.3
Class	1 sophomore 69 juniors 8 seniors	1.2 88.5 10.2
Professional Status	30 employed 48 unemployed	38.5 61.5

**Table 2 ijerph-19-01270-t002:** T1 (pre) to T2 (during) measurements of the POMS questionnaire and objectively measured steps per week.

Variables	*t*-Value	*p*-Value
Tension	0.197	0.845
Depression	1.376	0.173
Anger	0.621	0.536
Vigour	−7.455	≤0.001
Fatigue	−0.874	0.385
Confusion	−0.197	0.844
Total Mood Disturbance	2.255	0.027
Steps/wk	−8.259	≤0.001

**Table 3 ijerph-19-01270-t003:** Associations between Profile of Mood State and Objective Physical Activity pre- and during Lockdown.

Variables	Adjusted R^2^	Beta	*t*-Value	*p*-Value
T1 Tension	−0.010	0.055	0.481	0.632
T2 Tension	0.024	−0.0192	−1.707	0.092
T1 Depression	−0.013	0.001	0.007	0.994
T2 Depression	0.064	−0.277	−2.511	**0.014**
T1 Anger	−0.001	−0.111	−0.971	0.334
T2 Anger	0.042	−0.233	−2.091	**0.040**
T1 Vigor	−0.010	0.053	0.463	0.644
T2 Vigor	0.068	0.283	2.570	**0.012**
T1 Fatigue	−0.012	−0.032	−0.277	0.782
T2 Fatigue	0.033	−0.213	−1.901	0.061
T1 Confusion	−0.013	−0.021	−0.179	0.858
T2 Confusion	0.021	−1.84	−1.629	0.108
T1 TMD	0.044	−0.035	−0.302	0.764
T2 TMD	0.079	−0.302	−2.754	**0.007**

## Data Availability

The data is located on the primary investigators computers that are protected by password entry.

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
