# Peer review of "The Importance of Physical Activity to Augment Mood during COVID-19 Lockdown"

_ijerph, 2022, doi:10.3390/ijerph19031270_

Round 1

Reviewer 2 Report

The paper “The Importance of Physical Activity to Augment Mood during 2 COVID-19 Quarantine” examines the psychological and physical impact of the COVID pandemic. To this aim the authors collected healthy participants’ questionnaire and kinematics data before and during quarantine. The study is sound and timely.

There might be room for improving the discussion in order to better highlight the impact of the study within a consolidated and broader framework.

For instance, with respect to the cascade of psychological and behavioral effects triggered by the COVID pandemic, it has been shown that the negativity of the psychological effects of the quarantine was further modulated by personality traits, alexithymia, and resilience (Osimo et al. 2021, Frontiers in Psychology) and that these effects were also correlated with behavioral wellbeing such as emotional eating (Cecchetto et al. 2021, Appetite). Linking these previous studies to the present one could provide new perspectives on how to further ameliorate COVID-related assessments, e.g. taking into account the influence of physical activity on the relationship between lockdown, personality traits, and behavioral wellbeing.

Reviewer 3 Report

Thank you for the author's contribution to this study. I have the following recommendation to improve the manuscript:

  • Please add previous studies to the literature review about the topic of depression, anger, etc. since these variables were included in the analysis.
  • I recommend using a subchapter for the first part of the material and methods. (e.g., procedure)
  • When does Phase 2 take part?
  • Please report Cronbach-alpha reliabilities for POMS
  • Please add the minimum and maximum values of POMS
  • Please clarify in the measures that "steps" were analyzed as an objective value.
  • Line 122 "(mean SD)" is not clear where it belongs. I recommend using as the following: ...(Mean age=22.4; SD=3.1)...
  • Add a caption to the graphs in figure 1

Round 2

Reviewer 3 Report

Thank you for your authors contribution. All required changes have been made. I recommend for publishing.